# Transcriptome Analysis of Differentially Expressed mRNA Related to Pigeon Muscle Development

**DOI:** 10.3390/ani11082311

**Published:** 2021-08-05

**Authors:** Hao Ding, Yueyue Lin, Tao Zhang, Lan Chen, Genxi Zhang, Jinyu Wang, Kaizhou Xie, Guojun Dai

**Affiliations:** 1College of Animal Science and Technology, Yangzhou University, Yangzhou 225000, China; 15138343214@163.com (H.D.); lyy3078539326@163.com (Y.L.); chenlan9326@163.com (L.C.); gxzhang@yzu.edu.cn (G.Z.); jywang@yzu.edu.cn (J.W.); kzxie@yzu.edu.cn (K.X.); daigj@yzu.edu.cn (G.D.); 2Joint International Research Laboratory of Agriculture and Agri−Product Safety, Ministry of Education, Yangzhou University, Yangzhou 225000, China

**Keywords:** pigeon, skeletal muscle, development and growth, WGCNA, transcriptomics

## Abstract

**Simple Summary:**

The growth and development of skeletal muscle determine the meat production performance of pigeons and are regulated by complex gene networks. To explore the genes involved in regulating the growth and development of pigeon skeletal muscle, RNA sequencing (RNA−seq) was used to characterise gene expression profiles during the development and growth of pigeon breast muscle and identify differentially expressed genes (DEGs) among different stages. This study expanded the diversity of pigeon mRNA, and it was helpful to understand the role of mRNA in pigeon muscle development and growth.

**Abstract:**

The mechanisms behind the gene expression and regulation that modulate the development and growth of pigeon skeletal muscle remain largely unknown. In this study, we performed gene expression analysis on skeletal muscle samples at different developmental and growth stages using RNA sequencing (RNA−Seq). The differentially expressed genes (DEGs) were identified using edgeR software. Weighted gene co−expression network analysis (WGCNA) was used to identify the gene modules related to the growth and development of pigeon skeletal muscle based on DEGs. A total of 11,311 DEGs were identified. WGCNA aggregated 11,311 DEGs into 12 modules. Black and brown modules were significantly correlated with the 1st and 10th day of skeletal muscle growth, while turquoise and cyan modules were significantly correlated with the 8th and 13th days of skeletal muscle embryonic development. Four mRNA−mRNA regulatory networks corresponding to the four significant modules were constructed and visualised using Cytoscape software. Twenty candidate mRNAs were identified based on their connectivity degrees in the networks, including Abca8b, TCONS−00004461, VWF, OGDH, TGIF1, DKK3, Gfpt1 and RFC5, etc. A KEGG pathway enrichment analysis showed that many pathways were related to the growth and development of pigeon skeletal muscle, including PI3K/AKT/mTOR, AMPK, FAK, and thyroid hormone pathways. Five differentially expressed genes (LAST2, MYPN, DKK3, B4GALT6 and OGDH) in the network were selected, and their expression patterns were quantified by qRT−PCR. The results were consistent with our sequencing results. These findings could enhance our understanding of the gene expression and regulation in the development and growth of pigeon muscle.

## 1. Introduction

Pigeon meat is rich in nutrition, high in protein, low in fat, and high in medicinal value. In China, pigeon meat is called “animal ginseng” and is considered an advanced nourishment product that is increasingly favoured by consumers [1]. Meat production performance is an important index to measure the economic value of pigeons. However, the genetic improvement of the meat production performance of pigeons is relatively lagging in comparison with other poultry. The growth and development of skeletal muscle determine the meat production performance of pigeons [2]. Therefore, understanding the molecular regulation mechanism of pigeon skeletal muscle growth and development is an important prerequisite for improving meat production performance by molecular breeding technology [3].

The development of skeletal muscle is closely related to skeletal muscle cell differentiation. Myogenic regulatory factor family (MRF) members play essential roles in the process of skeletal muscle cell differentiation, including MyoD (myogenic determining factor), MyoG (myogenin), MRF4 (myogenic regulatory factor 4), and Myf5 (myogenic factor 5) [4]. However, the process of skeletal muscle growth and development involves multi−gene expression, signal transduction, and network regulation, and there are still a large number of regulatory factors to be identified.

RNA sequencing (RNA−seq) can explore the differences of gene types and expression levels at the overall level and directly link the changes of gene expression levels with phenotypic changes. In recent years, to explore the molecular mechanism of critical economic traits, considerable transcriptome studies on livestock and poultry were carried out using RNA−seq technology [5]. Xing et al. revealed essential genes related to muscle fat and abdominal fat deposition in chickens during the development process and identified 21 key genes in total, using RNA−Seq analysis [6]. Hu et al. identified several genes and pathways that may regulate skeletal muscle growth in the black Muscovy duck using RNA−seq [7]. RNA−seq technology has also been applied to pigeon transcriptome analysis. Wang et al. performed a comprehensive investigation into miRNA transcriptomes in livers across three pigeon developmental stages using RNA−seq and identified several vital target genes (e.g., TNRC6B, FRS2, PTCH1, etc.) of DE miRNA, which is closely linked to liver development [8]. Ye et al. compared the transcriptomes of muscle and liver tissues between squabs of two breeds to identify candidate genes associated with the differences in the capacity of fat deposition. A total of 27 genes were identified as a basis for further investigations to screen markers closely associated with intramuscular fat content and fatty acid composition in squabs [9].

However, the application of RNA−Seq technology in mining genes related to pigeon skeletal muscle growth and development has not been reported yet. The regulatory mechanisms of pigeon skeletal muscle growth and development remain poorly understood. Therefore, the present study aims to characterise gene expression profiles during pigeon muscle development and growth and identify key genes involved in this biological process. Our findings will contribute to a better understanding of the mechanisms by which genes regulate pigeon muscle development and growth.

## 2. Materials and Methods

### 2.1. Animal Ethics Statement

This study was performed following the Chinese guidelines for animal welfare, and the animal protocol was approved by the Animal Welfare Committee of Yangzhou University (permit number SYXK [Su] 2016–0020).

### 2.2. Experimental Animals

The Tarim pigeon is a local breed of semi−wild pigeons in Yeerqiang River Basin, which has strong disease resistance, high reproductive capacity and good stress resistance. It is mainly distributed in Hotan, Kashgar and Aksu. Tarim pigeons are a kind of meat and egg dual−purpose pigeon, which primarily feeds on various grains and can go out in groups.

### 2.3. Sample Preparation

Twelve pigeons were obtained from Wuxi Sanxiangan Agricultural Technology Development Co., Ltd. (Jiangsu, China). The fresh left breast muscle tissues were collected, immediately frozen in liquid nitrogen, and stored in liquid nitrogen until use. A total of 12 samples of four stages (3 replicates for each stage, 8 and 13 embryonic age, 1 and 10 day-olds) were collected.

### 2.4. Library Construction and RNA Sequencing

A total of twelve cDNA libraries were constructed with muscle tissues, and 3 μg total RNA per sample was used as the input material for a cDNA library. After total RNAs were extracted, rRNAs were removed, and then the enriched RNAs were fragmented into short fragments and reverse transcribed into cDNAs. Double−stranded cDNAs were synthesised by replacing dTTPs with dUTPs in the reaction buffer used in second−strand cDNA synthesis. The resulting double−stranded cDNAs were ligated to adaptors after being end−repaired and A−tailed. Then, uracil−N−glycosylase (UNG) was used to digest the second−strand cDNAs. The digested products were size selected by agarose gel electrophoresis, PCR amplified and sequenced by Gene Denovo Biotechnology Co. (Guangzhou, China) using Illumina HiSeq 4000 (150 bp paired−end).

### 2.5. mRNA Identification and Quantification

High−quality clean reads were obtained by removing reads containing adapters, reads consisting of all A bases, reads containing more than 10% of unknown nucleotides (N), and reads containing more than 50% of low quality (Q-value ≤ 20) bases. The high−quality clean reads were aligned against the ribosome RNA (rRNA) database using Bowtie2 [10] (version 2.2.8). The rRNA mapped reads then were removed. The remaining clean reads of each sample were then mapped to the *Columba livia* reference genome (https://www.ncbi.nlm.nih.gov/assembly/GCA_000337935.2/, 13 December 2017) by TopHat2 [11] (version 2.1.1) with default parameters, respectively. Fragments per kilobase of transcript per million mapped reads (FPKM) algorithm was used to quantify the expression levels of mRNAs.

### 2.6. Identification of Differentially Expressed Genes

The edgeR software was used to identify differentially expressed mRNAs. Genes with *p*-value < 0.05 and fold change ≥2 were considered significantly differentially expressed genes (DEGs). 

### 2.7. Co−Expression Network Analysis and Visualisation

Co−expression network analysis was analysed using the weighted gene co−expression network analysis (WGCNA) R package [12] based on the DEGs. The co−expressed modules were detected using the automatic network construction function blockwiseModules with power = 5. The hierarchical clustering of identified modules was conducted by applying the Dynamic Tree Cut algorithm. Each of these modules was summarised by its first principal component referred to as its eigengene, providing a single value for a module’s expression profile [13]. In order to identify modules associated with pigeon skeletal muscle development, eigengenes were correlated with the different stages of development and growth. Modules with an average eigengene−stage correlation of r > 0.4 were considered significantly associated with pigeon skeletal muscle development and growth. The top 1000 interactions of each significant module were exported and visualised using Gephi software (version 0.9.2) [14].

### 2.8. Functional Enrichment Analysis 

To explore the functions of the significant modules, genes in each module were submitted to Gene Ontology (GO) and Kyoto Encyclopedia of Genes and Genomes (KEGG) enrichment analysis. The GO biological process terms and KEGG pathways with *p* value  <  0.01 were considered significantly enriched GO terms and KEGG pathways.

### 2.9. qRT−PCR Confirmation of Differentially Expressed Genes

mRNA was reverse−transcribed respectively using HiScript Q RT SuperMix for qPCR (+gDNA wiper) (Vazyme, Nanjing, China) following the manufacturer’s recommendations. The mRNA forward primers were obtained commercially from Tsingke Biotechnology (Beijing, China) (Table 1). The ChamQ SYBR qPCR Master Mix (Low ROX Premixed) (Vazyme, Beijing, China) was used to perform the qRT−PCR procedure following the manufacturer’s recommendations. The reaction mixtures were added in a 96−well plate at 95 °C for 30 s, followed by 40 cycles of 95 °C for 10 s and 60 °C for 30 s, and then followed by a melting curve using an ABI 7500 Real−Time PCR System (Applied Biosystems, Foster City, CA, USA). All experiments were performed in triplicate for each biological repeat. Quantification of selected gene expression was performed using the comparative threshold cycle (2^−∆∆Ct^) method. The quantitative real−time PCR (RT−qPCR) results for all genes were statistically tested using the Student’s *t*-test (version SPSS 20.0).

## 3. Results

### 3.1. Quality Control

A total of 1,003,374,810 clean reads were obtained. The GC content of the clean reads across the 12 samples varied from 43.49% to 45.84%. The percentage of Q20 bases and Q30 bases for the clean reads were above 97.85% and 93.30%, respectively. After quality control, the percentage of high−quality clean reads of the 12 samples were above 98.89%, indicating that the sequencing data can be used for subsequent data analysis (Table 2).

### 3.2. Identification of Differentially Expressed Genes

A total of 30,900 transcripts (including 16,522 known transcripts and 14,378 new transcripts) were identified (Appendix A). Figure 1B showed the correlation between the samples based on gene expression measured using Pearson’s correlation coefficient. The results showed that the genes in pigeon skeletal muscle have developmental stage− and time−specific patterns. Samples of the embryonic stage (E8 and E13) are tightly clustered into the same subgroup, while the 1-day-old and 10-day-old samples were clustered into distinct subgroups, indicating varying gene expression of pigeon skeletal muscle between the embryonic and the growth stage.

A total of 11,311 genes were identified as DEGs among the four stages (*p* < 0.05, FC > 2) (Appendix A). The E8−D10 comparison group had the largest DEGs of 8242, including 5874 downregulated DEGs and 2368 upregulated DEGs. In contrast, the E8−E13 comparison group had the least number of DEGs of 1959, of which 1057 were downregulated, and 902 were upregulated (Figure 1A,C).

### 3.3. Weighted Gene Co−Expression Network Analysis

To investigate the relationship between DEGs and muscle development and growth, we performed a WGCNA analysis based on the expression of 11,311 DEGs. WGCNA categorised 11,311 DEGs into 12 modules with a soft power of 5 (Figure 2A,B). Among the 12 modules, black and brown were significantly correlated with day 1 and 10 stages, while turquoise and cyan were significantly correlated with embryonic development stages of days 8 and 13 (r > 0.4) (Figure 2C).

### 3.4. Visualisation of the Pigeon Muscle Development and Growth−Related Modules

To further explore candidate DEGs that regulate pigeon skeletal muscle growth and development, we visualised the four significant modules identified by WGCNA based on the top 1000 mRNA−mRNA interaction pairs in each module. Four mRNA−mRNA regulatory networks were constructed, corresponding to black, brown, cyan, and turquoise modules, respectively. The black, brown, cyan and turquoise modules contained 265, 308, 276, and 402 mRNAs, respectively (Figure 3).

### 3.5. Identification of Candidate Genes Related to Pigeon Muscle Development and Growth

We used Cytoscape software to calculate the degree of connectivity of the genes in the mRNA−mRNA network, and DEGs with high connectivity were considered to play an essential role in the network. In this study, the top five DEGs ranked by the degree of connectivity in each network were considered as hub genes. A total of 20 hub genes were identified (Table 3). The turquoise and cyan were development−specific modules, indicating that hub genes in turquoise and cyan modules might play potential roles in pigeon skeletal muscle development. The black and brown modules were significantly correlated to the 1−day−old and 10−day−old stages, suggesting that hub genes in the black and brown modules were involved in regulating the early growth stage of pigeon skeletal muscle.

### 3.6. GO and KEGG Pathway Enrichment Analysis

To further explore the function of DEGs in the four significant modules and the possible molecular mechanism by which DEGs regulated pigeon skeletal development and growth, we performed GO and KEGG pathway enrichment analysis on the DEGs in the black, brown, cyan and turquoise network, respectively. GO enrichment analysis showed that DEGs in the black module were significantly enriched in 91 biological processes (*p* < 0.01), such as cellular protein localisation, cellular macromolecule localisation and protein import (Figure 4A). DEGs in the brown module were significantly categorised into 47 biological processes (*p* < 0.01), including the generation of precursor metabolites and energy, the oxidation−reduction process and energy derivation by oxidation of organic compounds (Figure 4B). DEGs in the cyan module were assigned to 14 significantly enriched biological processes, such as muscle organ development, response to osmotic stress and striated muscle tissue development (Figure 4C). DEGs in the turquoise module were significantly categorised into 78 biological processes, including DNA−templated transcription, initiation and negative regulation of RNA metabolic process (Figure 4D). The significantly enriched GO molecular function and cellular component terms are listed in Appendix A. KEGG pathway enrichment showed that DEGs in the black module were enriched in signalling pathways such as PI3K−Akt signaling pathway and human papillomavirus infection (Figure 5A). DEGs in the brown module were enriched in signalling pathways such as thermogenesis and metabolic pathways (Figure 5B). DEGs in the cyan module were enriched in signalling pathways such as the rap1 signalling pathway and regulation of actin cytoskeleton (Figure 5C). DEGs in the turquoise module were enriched in signalling pathways such as viral carcinogenesis and microRNAs in cancer and RNA transport (Figure 5D).

### 3.7. qRT−PCR

In order to confirm the differentially expressed genes among the modules obtained by RNA−Seq, five differentially expressed genes (LAST2, MYPN, DKK3, B4GALT6 and OGDH) in the network were selected, and their expression patterns were quantified by qRT−PCR. The results are consistent with our sequencing results, highlighting the reliability of our sequencing data (Figure 6).

## 4. Discussion

Elucidating the molecular regulation mechanism of pigeon skeletal muscle growth and development is an essential prerequisite for using molecular breeding technology to improve pigeon meat production performance. However, no previous studies on identifying candidate genes and exploring mechanisms have been conducted in pigeon skeletal muscle development and growth to our best knowledge. 

One of the aims of this study was to characterise gene expression patterns during pigeon skeletal muscle development and growth at the mRNA level using RNA−seq. We found a significant difference in gene expression patterns between the development stage and the growth stage of pigeon skeletal muscle. It suggests distinct mechanisms of genes to regulate the development and growth in pigeon skeletal muscle. In the study of Li et al., co−expression analysis was used to determine that four modules were related to the specific growth stage of chicken breast muscle development [15].

Weighted gene co−expression network analysis is a systems biology method for describing the correlation patterns among genes across microarray samples. WGCNA can be used for finding clusters (modules) of highly correlated genes, for summarising such clusters using the module eigengene or an intramodular hub gene, for relating modules to one another and to external sample traits (using eigengene network methodology), and for calculating module membership measures [16]. For example, Chen et al. used WGCNA to identify some key genes in abdominal aortic aneurysms [17]. Li et al. analysed the gene co−expression network and functional modules of chicken breast muscle at different developmental stages [15]. Zhang et al. used WGCNA to analyse microbial and metabolic data sets [18]. WGCNA is widely used in biological analysis. 

We constructed the gene regulatory network in each module to screen further the key mRNA regulating the growth and development of pigeon skeletal muscle. In the four modules, according to the degree, a total of 20 candidate key mRNAs were identified, including the tumoural expression of the ABC transporter A8b (ABCA8b), TCONS−00004461, von Willebrand factor (VWF), 2−oxoglutarate dehydrogenase (OGDH), TGF−beta induced factor homeobox 1 (TGIF1), Dickkopf homolog 3 (DKK3), glutamine−fructose−6−phosphate transaminase 1 (GFPT1) and replication factor C 5 (RFC5). Many of these mRNAs are involved in the regulation of muscle growth and development. For example, the kinesin family member 1C (KIF1C) is known to regulate podosomes, actin−rich adhesion structures that remodel the extracellular matrix during physiological processes [19], Gache et al. found that KIF1C affects the differentiation of muscle cells [20]. Myopalladin (MYPN) is a striated muscle−specific, immunoglobulin−containing protein located in the Z−line and I−band of the sarcomere as well as the nucleus. MYPN promotes skeletal muscle growth by activating the serum response factor (SRF) pathway in muscle [21]. DKK3 is a stress−induced, renal tubular epithelia−derived, secreted glycoprotein (molecular mass, 38 kDa) [22] that induces tubulointerstitial fibrosis through its action on the canonical Wnt/*β*−catenin signalling pathway [23]. DKK3 is a divergent member of the DKK protein family. Wilde searched for new genes related to specific muscle types and found that the expression of DKK3 in mouse quadriceps was higher than that in other tissues [24]. Yin et al. confirmed that Dkk3 is the key secretory factor of muscle production [25]. The remaining key genes, such as Abca8b, TCONS−00004461, VWF, OGDH, TGIF1, Gfpt1 and RFC5 have not been reported in muscle growth and development.

We analysed the mRNA of black, brown, cyan and turquoise modules by GO and KEGG enrichment to explore the possible mechanism of mRNA regulation of pigeon skeletal muscle growth and development. Go enrichment showed that genes in the cyan module were enriched in many biological processes related to the growth and development of skeletal muscle, including muscle organ development, striated muscle tissue development, muscle tissue morphogenesis, muscle tissue development, muscle organ morphogenesis, muscle structure development and embryo development. It suggests that the mRNA in cyan may play an important regulatory role in the differentiation and development of pigeon skeletal muscle cells. The mRNA in black, brown and turquoise is significantly enriched in biological processes, such as biological process regulation, cell process regulation and tissue morphogenesis, indicating that the above biological processes may be related to the growth of pigeon skeletal muscle.

Pigeon muscle growth is a complex process affected by multiple genes and regulated by multiple pathways. Many pathways of mRNA enrichment in the four modules are related to muscle growth and development. GRP94 promotes muscle differentiation by inhibiting PI3K/AKT/mTOR signalling pathway [26]. AMPK plays an important role in controlling skeletal muscle development and growth due to its effects on anabolism and catalytic cellular processes [27]. Quach [28] found that FAK regulates the expression of a set of muscle−specific genes specifically involved in myoblast fusion during early myogenic differentiation, including β1D integrins. Cui et al. found that ISLR can relieve skeletal muscle atrophy and prevent muscle cell apoptosis through the IGF/PI3K/AKT−FOXO signalling pathway [29]. Ge et al. [30] studied the important effect of the interaction between bioactive nanomaterials and muscle cells on enhancing skeletal muscle tissue regeneration and found that bioactive nanomaterials can activate the p38+MAPK signalling pathway and enhance myogenic differentiation. Jaafar’s research found that the size of muscle cells is positively regulated by phospholipase D (PLD). Phospholipase D regulates the size of skeletal muscle cells by activating mTOR signals, thereby affecting muscle growth and development. Ambrosio [31] found that the proliferation and differentiation of skeletal muscle stem cells are completely dependent on the role of the thyroid hormone. Ribosome biogenesis in eukaryotes has become an important regulator of skeletal muscle growth and maintenance [32]. 

## 5. Conclusions

Our research results identified 11,311 DEGs and four co−expressed gene modules associated with the development and growth of pigeon skeletal muscle. Twenty candidate genes were identified, including Abca8b, TCONS−00004461, VWF, OGDH, TGIF1, DKK3, Gfpt1 and RFC5. Our results profiled gene expression of pigeon skeletal muscle samples at different stages and identified DEGs, which will contribute to our understanding of the mechanisms underlying the development and growth of pigeon skeletal muscle.

## Figures and Tables

**Figure 1 animals-11-02311-f001:**
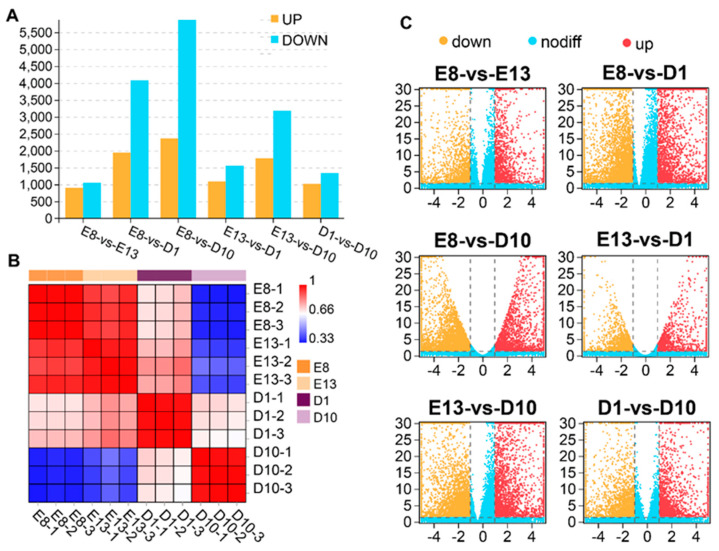
Gene expression pattern analysis. (**A**) Histogram of DEGs among the four stages. Blue represents the number of upregulated genes, and yellow represents the number of downregulated genes. (**B**) Heatmap of correlation between samples based on gene expression. The colours from blue to red represent low to high correlation. (**C**) Volcano plot of differential gene expression analysis. Upregulated genes are marked red, downregulated genes in yellow, and nonsignificant genes in blue, respectively.

**Figure 2 animals-11-02311-f002:**
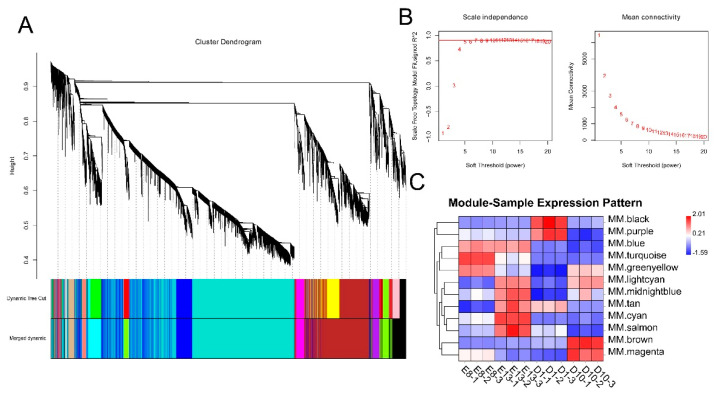
WGCNA analysis of DEGs. (**A**) Clustering dendrogram of DEGs with dissimilarity based on the topological overlap. Dynamic tree cutting was applied to identify modules by dividing the dendrogram at significant branch points. Modules are displayed with different colours in the horizontal bar immediately below the dendrogram. (**B**) Determination of soft−thresholding power for WGCNA analysis. The graph indicates that soft thresholding power above 5 meets scale−free topology above 0.85. (**C**) Heatmap of the correlation between samples and gene modules. Each row corresponds to a module eigengene, each column to a sample. Each cell contains the corresponding correlation value. Red and blue colours represent positive and negative correlations, respectively. The darker the colour, the higher the correlation.

**Figure 3 animals-11-02311-f003:**
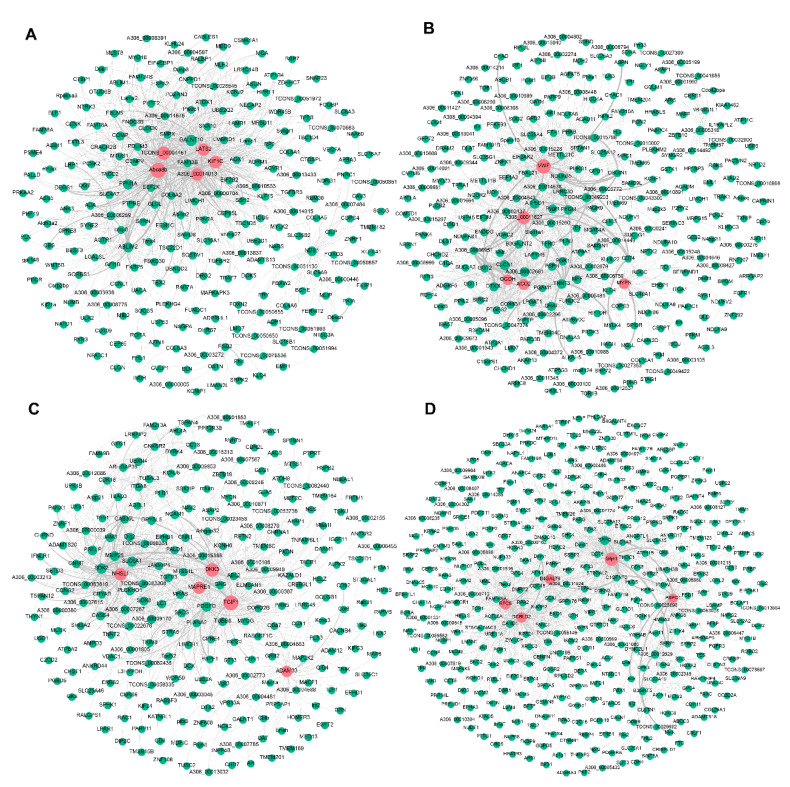
Visualisation of the four modules related to pigeon muscle development and growth. Networks (**A**–**D**) represent visualisations of gene interactions of black, brown, cyan and turquoise modules, respectively. The node size indicates the node degree. The line thickness indicates the strength of the correlation. Red nodes represent hub DEGs in each module.

**Figure 4 animals-11-02311-f004:**
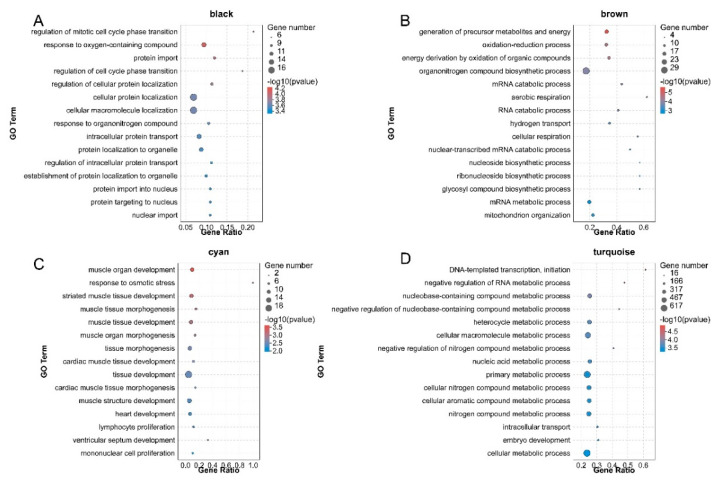
Bubble plots of the GO enrichment analysis of DEGs in the four pigeon skeletal muscle−related modules. (**A**–**D**) represent GO enrichment analysis of DEGs in black, brown, cyan and turquoise modules. Colours from red to blue indicate *p*-value from small to large. The bubble size indicates the number of DEGs included in each biological process.

**Figure 5 animals-11-02311-f005:**
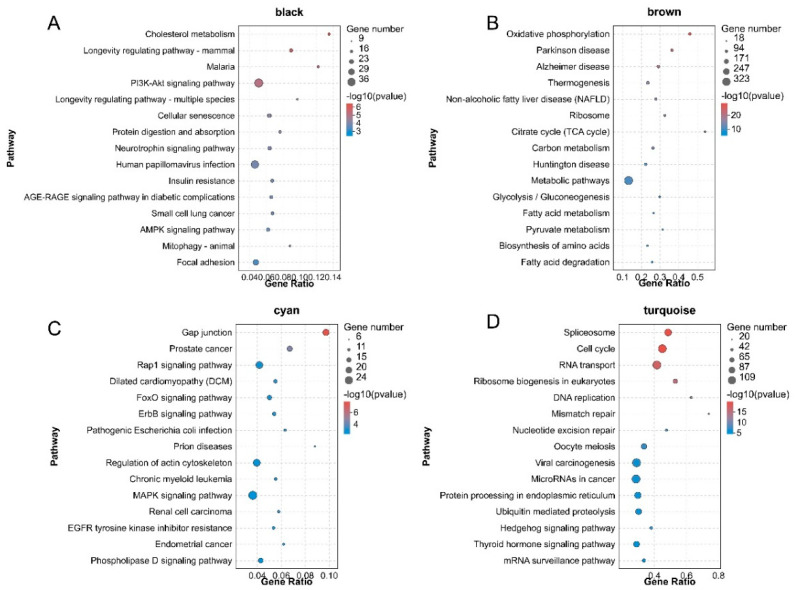
Bubble plots of the KEGG pathway enrichment of DEGs in the four pigeon skeletal muscle−related modules. (**A**–**D**) represent KEGG pathway enrichment analysis of DEGs in black, brown, cyan and turquoise modules. Colours from red to blue indicate *p*-value from small to large. The bubble size indicates the number of DEGs included in each biological process.

**Figure 6 animals-11-02311-f006:**
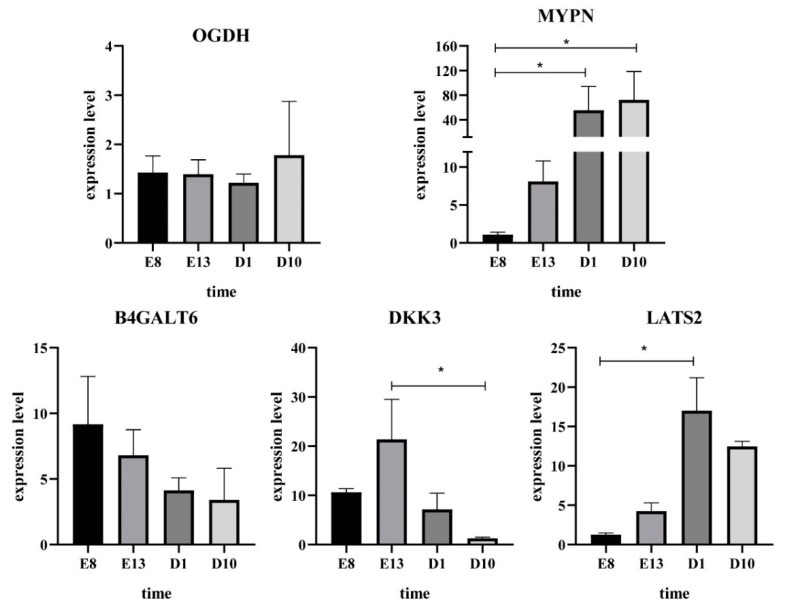
Validation of hub genes using RT−qPCR. “*” was considere dsignificant difference (*p* < 0.05).

**Table 1 animals-11-02311-t001:** Primers for RT−qPCR.

Genes	Primer Sequences
LAST2	F: GCGATCAGGAGATGGCTGTT R: GGCCTGCGGGTTACGTTATT
MYPN	F: TGGTAGGAATCCCAGCACCT R: AACGTCCCGTATCCTCCTCA
DKK3	F: AGTGGCTTGATCTGCCAACC R: TTGCGCACTTCCTGGATGAC
B4GALT6	F: CTGTTGGTTTCAAATGCTCTGG R: GGATAATCTGTGCCATTCAGTGT
OGDH	F: CGCTCATCAGGGCGTATCAG R: CCGTAAAAGCCAACGTTTGGAG
*β*−actin	F: CAGCCATCTTTCTTGGGTAT R: CTGTGATCTCCTTCTGCATCC

**Table 2 animals-11-02311-t002:** Quality control of the clean reads.

Sample	Clean Reads	High Quality Clean Reads	Q20 (%)	Q30 (%)	GC (%)	Mapping Ratio
E8−1	75,063,382	74,277,032 (98.95%)	10,791,351,832 (98.02%)	10,322,706,924 (93.76%)	4,800,811,241 (43.61%)	84.58%
E8−2	93,609,304	92,580,780 (98.9%)	13,424,507,726 (97.85%)	12,802,556,480 (93.32%)	6,026,072,771 (43.92%)	84.68%
E8−3	85,195,516	84,385,794 (99.05%)	12,293,074,970 (98.18%)	11,789,502,460 (94.16%)	5,445,575,947 (43.49%)	85.32%
E13−1	79,976,144	79,228,386 (99.07%)	11,539,085,039 (98.13%)	11,056,258,233 (94.02%)	5,177,072,410 (44.03%)	84.58%
E13−2	90,672,942	89,669,830 (98.89%)	12,993,101,531 (97.85%)	12,388,401,793 (93.30%)	5,922,761,289 (44.60%)	84.59%
E13−3	77,454,184	76,687,330 (99.01%)	11,162,607,212 (98.12%)	10,693,691,406 (94.00%)	5,028,023,238 (44.20%)	85.11%
D1−1	91,584,814	90,719,896 (99.06%)	13,208,290,831 (98.12%)	12,652,127,189 (93.99%)	5,870,226,127 (43.61%)	84.72%
D1−2	71,920,124	71,242,502 (99.06%)	10,385,891,415 (98.17%)	9,958,238,118 (94.13%)	4,628,302,383 (43.75%)	84.71%
D1−3	79,619,168	78,778,536 (98.94%)	11,436,594,412 (97.93%)	10,924,091,792 (93.54%)	5,102,169,115 (43.69%)	83.94%
D10−1	91,243,706	90,338,000 (99.01%)	13,131,770,227 (98.06%)	12,570,353,487 (93.87%)	6,042,016,224 (45.12%)	83.35%
D10−2	79,347,520	78,636,042 (99.1%)	11,461,080,321 (98.18%)	10,989,484,007 (94.14%)	5,350,951,502 (45.84%)	83.96%
D10−3	87,688,006	86,757,674 (98.94%)	12,614,551,950 (98.04%)	12,072,006,201 (93.82%)	5,762,097,126 (44.78%)	83.90%

**Table 3 animals-11-02311-t003:** Hub genes in the four modules.

Network	Hub mRNA	Degree
black	Abca8b	176
	TCONS−00004461	137
	A306−00014013	119
	KIF1C	114
	LATS2	111
brown	VWF	123
	A306−00011627	98
	OGDH	59
	ACO2	65
	MYPN	59
cyan	TGIF1	125
	DKK3	104
	NHSL1	102
	MAPRE1	81
	SDK2	68
turquoise	Gfpt1	229
	RFC5	143
	DCBLD2	130
	B4GALT6	105
	PSPC1	105

## Data Availability

The data that support the findings of this study are available from the corresponding author upon reasonable request.

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
