# Peer review of "Transcriptome Analysis of Differentially Expressed mRNA Related to Pigeon Muscle Development"

_animals, 2021, doi:10.3390/ani11082311_

Round 1

Reviewer 1 Report

The work of Ding et al. contributes to the knowledge of the processes of regulation of growth and development of skeletal muscle, which may have benefits in the medium term to improve the performance of meat production through molecular reproduction technology. 
There are some points of the methodology that are not clear to me regarding its implementation.

Major revision

1. What type of statistical test was used for the enrichment analysis? Or what software was he using? 

2. Describe in a more explicit way which is the reference that is used in the RT-qPCR to quantify the expression of the analyzed genes. Did you use a baseline expression reference gene? 

3. Why are the molecular processes of GO not described? 

4. What is the explanation or the correlation between the development and growth of skeletal muscle and the KEGG pathways of Human papillomavirus infection, malaria, AGE-RAGE, prostate cancer, pathogenic E. coli infection, chronic myeloid leukemia, renal cell carcinoma, endometrial cancer, Alzheimer disease, Parkinson disease, Huntington disease, viral carcinogenesis shown in Figure 5? 

Minor revision

Line 21: Change "regulate" by "modulate"
Line 22:  Did you mean "unknown" or "known"
Line 27: Change "Balck" by "Black"
Line 30-31: Change "significnat" by "significant"
Line 56: The names is "muscle reguLatory factor 4" or "muscle regulatory factor 4" or "Myogenic regulatory factor 4".
Line 116: Was the analysis with Top HAt performed with RNA or with mRNA? 
Line 128: Eliminated one "u" in "conduucted"
Line 150: Is 945°C correct?
Line 152: Change "repea" by "repeat"
Line 162: Percentage of high-quality clean reads of the 12 samples were above 98.89% (Table 2)
Line 230: Change "Go" by "GO".
Line 270-271: I don't understand this paragraph "here are few reports on the related genes regulating the muscle growth mechanism of pigeons"
Line 285: Change "Analyzed" by "analyzed".
Line 312: Change "balck" by "black"
Line 312: Change "Turquise" by "turquise"
Line 313: Change "go" by "GO".
Line 313: Change "Go" by "GO"
Line 314: Change "Cyan" by "cyan".
Line 317: Change "Cyan" by "cyan".
Line 319: Homogenize the names of the colors with upper or lower case throughout the text.

Author Response

Reviewer 1

Major revision

  1. What type of statistical test was used for the enrichment analysis? Or what software was he using? 

Response: The enrichment analysis was performed using the OmicShare tools,a free online platform for data analysis (www.omicshare.com/tools). Significantly enriched GO terms and KEGG pathways were defined by the hypergeometric test. The GO biological process terms and KEGG pathways with P value < 0.01 were considered as significantly enriched GO terms and KEGG pathways, respectively. This has been added to the manuscript.

  1. Describe in a more explicit way which is the reference that is used in the RT-qPCR to quantify the expression of the analyzed genes. Did you use a baseline expression reference gene? 

Response: This has been added to section 2.9.

  1. Why are the molecular processes of GO not described? 

Response: Thanks for your comment. In our study, we only displayed the biological process results of GO enrichment analysis. There are two main reasons: Firstly, the development and growth of skeletal muscle are regulated by many biological processes such as myoblast proliferation, fusion, and myotube differentiation. However, the biological processes involved in skeletal muscle development and growth were rarely reported. Thus, our study mainly focused on exploring the biological processes rather than the molecular function and cellular component terms related to the development and growth of pigeon skeletal muscle. Secondly, we performed GO enrichment analysis with DEGs from four modules. If the significantly enriched biological processes, molecular function, and cellular component terms are shown at the same time, it will make the manuscript too long and difficult to read and understand.

Moreover, we uploaded the significantly enriched molecular function and cellular component as supplementary material in Table S3.

  1. What is the explanation or the correlation between the development and growth of skeletal muscle and the KEGG pathways of Human papillomavirus infection, malaria, AGE-RAGE, prostate cancer, pathogenic E. coli infection, chronic myeloid leukemia, renal cell carcinoma, endometrial cancer, Alzheimer disease, Parkinson disease, Huntington disease, viral carcinogenesis shown in Figure 5? 

Response: We also found that the DEGs in the four modules were significantly enriched in disease-related pathways. We checked the KEGG pathway enrichment results and found that this is mainly because some DEGs were enriched in multiple KEGG pathways simultaneously. For example, in the black module, the LRP1 gene was enriched in the cholesterol metabolism pathway, and it was simultaneously involved in the malaria pathway.

Minor revision

Line 21: Change “regulate” by “modulate”

Response: The “regulate” has been changed to “modulate”.

Line 22:  Did you mean “unknown” or “known”

Response: “known” has been changed to “unknown”.

Line 27: Change “Balck” by “Black”

Response: “Balck” has been changed to “Black”.

Line 30-31: Change “significnat” by “significant”

Response: “significnat” has been changed to “significant”.

Line 56: The names is “muscle reguLatory factor 4” or “muscle regulatory factor 4” or “Myogenic regulatory factor 4”.

Response: It has been changed to “Myogenic regulatory factor 4”.
Line 116: Was the analysis with Top HAt performed with RNA or with mRNA?

Response: It should be clean reads without rRNA sequences. We have revised related sentences to make them easy to understand. (The high-quality clean reads were aligned against the ribosome RNA (rRNA) database using Bowtie2 (version 2.2.8). The rRNA mapped reads then were removed. The remaining clean reads of each sample were then mapped to the Columba livia reference genome by TopHat2 (version 2.1.1) with default parameters, respectively.)

Line 128: Eliminated one “u” in “conduucted”

Response: It has been corrected to “conducted”.
Line 150: Is 945°C correct?

Response: It has been corrected to 95°C.
Line 152: Change “repea” by “repeat”

Response: The “repea” has been changed to “repeat”.
Line 162: Percentage of high-quality clean reads of the 12 samples were above 98.89% (Table 2)

Response: It has been corrected.

Line 230: Change “Go” by “GO”.

Response: The “Go” has been changed to “GO”.

Line 270-271: I don’t understand this paragraph “here are few reports on the related genes regulating the muscle growth mechanism of pigeons”

Response: This sentence is redundant and has been deleted.

Line 285: Change “Analyzed” by “analyzed”.

Response: It has been changed.
Line 312: Change “balck” by “black”

Response: It has been changed.
Line 312: Change "Turquise" by "turquise"

Response: It has been chanegd.
Line 313: Change “go” by “GO”.

Response: It has been changed.
Line 313: Change “Go” by “GO”

Response: It has been chanegd.
Line 314: Change “Cyan” by “cyan”.

Response: It has been chanegd.
Line 317: Change “Cyan” by “cyan”.

Response: It has been chanegd.
Line 319: Homogenize the names of the colors with upper or lower case throughout the text.

Response: Changes have been made at the corresponding location.

Reviewer 2 Report

In this paper, the authors report transcriptome analysis of pigeon skeletal muscle at several stages of development, and conduct differential gene analysis as well as network analysis of the results. This paper contributes to our understanding of genes expressed during pigeon muscle development, however, a number of revisions should be made before the article is considered for publication.

Major concerns:

  1. In section 2.5, the authors indicate that they mapped reads to the chicken genome. They should map reads to the pigeon genome instead, since one is publicly available and annotated at https://www.ncbi.nlm.nih.gov/assembly/GCA_000337935.2/
  2. Related to the above concern, the authors should report in section 3.1 or 3.2 what percentage of reads were successfully mapped to a transcript. Mapping to an organism of a different genus would likely cause a significant number of unaligned reads due to sequence divergence.
  3. The authors should describe what type of sequencing was performed; single read or paired-end, and the number of nucleotides per read.
  4. The authors should indicate what mapping parameters were used by TopHat2.

Minor concerns:

  1. “Black” is misspelled on lines 27, 312, and 319.
  2. “Turquoise” is misspelled on lines 28-29.
  3. “GO” should be capitalized both times it is written on line 313.
  4. On lines 314-315, “Cyan. Including” should be re-written as “Cyan, including”.

Author Response

Reviewer 2

Major concerns:

  1. In section 2.5, the authors indicate that they mapped reads to the chicken genome. They should map reads to the pigeon genome instead, since one is publicly available and annotated at https://www.ncbi.nlm.nih.gov/assembly/GCA_000337935.2/

Response: The "Gallus gallus" has been changed to "Columba livia". The corresponding sentence has revised to "The remaining clean reads of each sample were then mapped to the Columba livia refer-ence genome (https://www.ncbi.nlm.nih.gov/assembly/GCA_000337935.2/) by TopHat2 (version 2.1.1) with default parameters, respectively."

  1. Related to the above concern, the authors should report in section 3.1 or 3.2 what percentage of reads were successfully mapped to a transcript. Mapping to an organism of a different genus would likely cause a significant number of unaligned reads due to sequence divergence.

Response: This has been added to section Table.2.

  1. The authors should describe what type of sequencing was performed; single read or paired-end, and the number of nucleotides per read.

Response: Sequencing was performed using paired-end 150 base reads. This has been added to section 2.4.

  1. The authors should indicate what mapping parameters were used by TopHat2.

Response: The default parameters were used. This has been added to section 2.5.

Minor concerns:

  1. "Black" is misspelled on lines 27, 312, and 319.

Response: The misspelled "Black" have been corrected.

  1. "Turquoise" is misspelled on lines 28-29.

Response: It has been corrected.

  1. "GO" should be capitalized both times it is written on line 313.

Response: It has been corrected.

  1. On lines 314-315, "Cyan. Including" should be re-written as "Cyan, including".

Response: Changes have been made at the corresponding location.

Round 2

Reviewer 2 Report

The authors have addressed my concerns.